# Long-Term Year-Interval Effect of Continuous Maize/Soybean Intercropping on Maize Yield and Phosphorus Use Efficiency

**DOI:** 10.3390/plants14071060

**Published:** 2025-03-29

**Authors:** Munir Ahmad, Tilei Zhao, Harun Gitari, Hongmin Zhao, Li Tang

**Affiliations:** 1College of Plant Protection, Yunnan Agricultural University, Kunming 650201, China; muniraup@gmail.com; 2College of Resources and Environmental Science, Yunnan Agricultural University, Kunming 650201, China; 3Department of Agricultural Sciences and Technology, Kenyatta University, Nairobi 00100, Kenya

**Keywords:** maize/soybean intercropping, yield sustainability, agronomic efficiency, recovery efficiency, partial factor productivity

## Abstract

The increasing global food demand, the degradation of one-third of agricultural land, and climate change pose significant threats to food production. Maize and soybean intercropping can enhance yields and land use efficiency, yet the year-interval effects of continuous intercropping on yield, yield sustainability, and phosphorus use efficiency (PUE) remain unclear. This study evaluates the effects of continuous maize/soybean intercropping over year intervals on yield, PUE, and sustainability. A seven-year field trial (2017–2023) was conducted on acidic soil, comparing two cropping systems: maize monocropping and maize intercropped with soybean. The results showed that continuous maize/soybean intercropping outperformed maize monocropping across all year intervals. Over the first, third, and seventh year intervals, maize yield increased by 37%, 35%, and 58%, respectively, with a 55% average increase over the seven years. Intercropping also enhanced P use efficiency, as evidenced by AE, RE, PFP, and CPF increases. In the first year, AE, PFP, RE, and CPF increased by 48%, 37%, 27%, and 16%, respectively; after the third year interval, these metrics improved by 40%, 35%, 26%, and 14%; and after the seventh year interval, they rose by 60%, 58%, 24%, and 10.5%. The average AE, RE, and PFP increases over seven years were 53%, 52%, and 27%, respectively, while CPF increased by 13%. The SEM analysis further confirmed the substantial impact of the seventh year intercropping interval on maize yield, sustainability, and PUE. This study demonstrates that continuous maize/soybean intercropping can enhance maize yield, PUE, and sustainability, with the seventh year interval offering the most pronounced benefits. These findings provide valuable insights for improving food security and nutrient management challenges.

## 1. Introduction

Agricultural production needs to be increased over time to satisfy the increasing global food demand [1,2]. However, the increased use of chemical fertilizer in the long term and the extensive sole cropping system harm the agricultural ecosystem and environment [3,4,5,6,7]. Therefore, the cropping system must be modified to guarantee worldwide food availability and ecological conservation. In this context, incorporating crop diversity into agricultural practices could boost crop yields, sustainable agriculture production, and resilience to environmental changes [8,9,10,11,12]. Intercropping entails co-cultivation of two or more identical or different crop species in proximity at the same or different times, thereby increasing crop yield and sustainability [13,14,15]. The increased yield in intercropping is ascribed to complementary ecological niches, disease control, and efficient utilization of natural resources such as nutrients, sunlight, and land [16,17,18,19,20]. Intercropping also improves plant nutrient uptake, use efficiency, and soil fertility through facilitative root interactions, legume N2 fixation, rhizosphere alteration (soil physio-chemical properties and enzymatic activity), root exudation, rich beneficial microbes, and some unknown mechanisms [14,19,21]. Intercropping is a key cropping practice that enhances phosphorus use efficiency in soil [22,23]. A substantial body of research has established that effective intercropping practices can enhance phosphorus uptake and utilization [24,25,26,27]. For instance, Zhou, Su [15] conducted research and a global meta-analysis and stated that maize/pea intercropping enhances productivity and phosphorus uptake efficiency by up to 13% against monocropping. Similarly, another global meta-analysis highlighted the potential of cereal/legume intercropping, which greatly elevated land use and phosphorus efficiency and produced an enhanced net phosphorus uptake of 3.67 kg P ha^−1^. Additionally, Ma, Yu [28] conducted a maize/alfalfa intercropping study and observed the alfalfa-positive complementary effect, which enhanced the phosphorus use efficiency and uptake by 53% and 61% compared to monocropping. Zhou, Su [15] conducted a two-year study in the subtropical area of southwestern China, demonstrating that maize/soybean mixtures under different phosphorus gradients greatly increased the phosphorus use efficiency by 43 to 74%. Maize/soybean intercropping is a widespread planting practice that increases relative grain yields and P uptake due to intercrop facilitative interactions and better resource use [15,29,30].

In China, maize/soybean intercropping is the most widespread intercropping system. This system has received attention due to its crop production and agricultural sustainability [15,31,32,33]. For instance, intercropping maize with soybean boosted the yield by 22% due to changes in soil phytochemicals such as organic matter, total nitrogen, and aggregate stability [14]. Studies have shown that long-term (≥10 years) maize/soybean intercropping improves land utilization, nutrient uptake, and crop production [34,35,36,37]. Similarly, short-term (≤3 years) studies on maize/soybean intercropping systems suggest yield increases of 56% and 60% at nitrogen fertilizer levels [19,38]. Similarly, Zhou, Su [15] investigated the impact of maize//soybean intercropping on maize yield stability under different phosphate application rates in red soil. The results showed a significant year-to-year increase in maize yield, with improvements observed under varying phosphorus rates. This suggests that intercropping substantially increases crop yield and stability and improves soil fertility and productivity compared to intensive sole-cropping systems.

To the extent of our understanding, most intercropping research has focused on short-term periods of studies. Similarly, several studies have investigated the effect of combining cereals and legumes on the mobilization and transformation of the insoluble form of P into the available form [15,39]. However, the impact of continuous intercropping on the maize grain yield, yield sustainability, and phosphorus use efficiency over long-term year intervals remains elusive. Thus, experiments were established to investigate the impact of continuous intercropping of maize with soybean on crop grain yield, sustainability, and phosphorus use efficiency over year intervals. We hypothesized that continuous maize/soybean intercropping in year intervals would promote phosphorus use efficiency, improving yield sustainability, grain yield, and the complementary interaction between maize and soybean.

## 2. Results

### 2.1. The Year-Interval Effects on Maize Yield and Yield Sustainability Under Maize/Soybean Continuous Intercropping

#### 2.1.1. The Year-Interval Effects on Yield and Sustainability

Our year intervals of continuous intercropping field trials revealed an increasing trend in maize grain yield across all the experimental years. We observed that maize yield was substantially affected by the intercropping system and phosphate application throughout the experimental period (Figure 1a). The yield increase in maize intercropping for seven years on average was 55%. Compared to the seven-year average, in the first year of the experiment, in 2017, maize intercropping showed an increase of 37%. Similarly, subsequent year intervals showed considerable yield advantages for maize intercropping. In the third year interval, the maize intercropping overyielding advantage was 35%. In the seventh year interval, the evident impact of intercropping on maize yield became more apparent, demonstrating the highest yield increase of 58% (Figure 1a). We performed a three-factor ANOVA to clarify our findings. We discovered that phosphorus application, year, and cropping systems greatly affected maize grain production, with a significant interaction effect (*p* < 0.01) from 2017 to 2023 (Table 1).

The year-interval experiment found a substantial impact of the cropping pattern and phosphorus fertilizer on maize yield SYI across the trial years (*p* < 0.05) (Figure 1b). In the seven-year average, the maize intercropping substantially boosted the SYI by 15%. Compared to the seven-year average SYI value, the lowest increase was recorded in the first year of the experiment in 2017 at 13% (Figure 1b). Similarly, the third year interval demonstrated an increase of 14%, while the subsequent seventh year interval revealed the highest increase in SYI of 20%.

#### 2.1.2. The Year-Interval Effects on Actual Yield Loss and Yield Stability

The actual yield loss (AYL) refers to the proportional yield gain or loss of intercrops compared to their corresponding sole crops. Across the experimental years from 2017 to 2023, intercropping maize AYL results exceeding 0 (AYLm ˃ 0) demonstrated the overyielding advantage of intercropping. The lowest increase in AYL values was recorded in the initial year of the experiment in 2017, compared to the seven-year average AYL values (Figure 2a). Similarly, in the third year interval in 2019, maize intercropping sustained its yield gain advantage and increased the AYL values. The subsequent seventh year interval in 2023 showed the highest AYL values compared to the seven-year average AYL values (Figure 2a).

The stability index (SI) is used broadly to compare and measure the crop’s yield variability over the years. A lower SI value suggested less yield variability, meaning more stability, whereas a higher SI value indicated lower yield stability. Across all the year intervals, we observed that cropping patterns, the conventional phosphate application rate, and their intra-interaction with the year enormously impacted the maize yield stability (*p* < 0.05). Monocropping maize exhibited SI values of 30%, 28%, 40%, and 69% higher than maize intercropping across the first year of 2017, the third year of 2019, the seventh year of 2023, and the average of seven years, respectively (Figure 2b).

### 2.2. The Year-Interval Effects on Phosphorus Use Efficiency Under Maize/Soybean Continuous Intercropping

#### 2.2.1. The Year-Interval Effects on the Contribution of Phosphate Fertilizer

The two-way ANOVA revealed that the contribution of the phosphate fertilizer rate (CPF) was significantly affected by the cropping patterns and the year (*p* < 0.05) over the experimental years. The maize’s CPF indicated an increasing trend under intercropping (Figure 3a). In the initial year of the experiment, in 2017, under the conventional rate of phosphorus fertilizer application, a 16% increase in CPF was observed under the maize intercropping. Similarly, in the third year interval in 2019, the seventh year interval in 2023, and the seven-year average, 14%, 10.5%, and 13% increases in the CPF were recorded, respectively. Compared to 2017, the third year interval in 2019 showed a slight increase in CPF of 3% in intercropping. In the subsequent seventh year interval, a notable rise in CPF was noted in 2023 for maize intercropping, with an increase in the phosphorus fertilizer contribution rate of 32% for intercropping maize. The seven-year average increase in the CPF was 12% for maize intercropping.

#### 2.2.2. The Year-Interval Effects on the Phosphate Use Efficiency

The agronomic efficiency of phosphate fertilizer (AE), the apparent recovery efficiency of phosphate fertilizer (RE), and partial factor productivity (PFP) are all critical indicators used to evaluate the efficiency of phosphate fertilizer application from different angles. The AE represents the increase in crop yield with each additional 1 kg of phosphate fertilizer (P_2_O_5_) application. The RE represents the proportion of fertilizer nutrients taken up by crops to the amount of fertilizer applied, which is an important index to reflect the uptake and utilization of fertilizer. The PFP reflects the comprehensive effect of the soil nutrient level and phosphate fertilizer application amount.

Across the experimental years from 2017 to 2023, the two-factor ANOVA findings indicated that the year, cropping pattern, and their interaction significantly affected AE, RE, and PFP (Figure 3b–d). The maize intercropping system gradually increased AE, RE, and PFP with the progression of planting years. All the trial years of field experiments showed that intercropping could significantly increase the AE, RE, and PFP under the conventional phosphate application rate. Compared with monoculture, intercropping maize substantially boosted AE by 48%, 40%, 60%, and 53% in 2017, the third year interval in 2019, the seventh year interval in 2023, and the average of seven years, respectively. Similarly, increased values were observed for the RE, by 37%, 35%, 58%, and 52%, respectively. In addition, the PFP revealed considerable increases of 27%, 26%, 24%, and 27%, respectively. In terms of year intervals, compared to the seven-year average data, the seventh year interval in 2023 showed the highest increases for AE, RE, and PFP (Figure 3b–d).

### 2.3. Structural Equation Model Analysis of the Year-Interval Effects Under Maize/Soybean Continuous Intercropping

Structural equation model analysis was performed on first, third, and seventh year-interval data to examine the effects of different variables on maize yield, AE, RE, PFP, and their co-relation. In the first year of the experiment, in 2017, the structural equation model (SEM) found no direct impact of phosphorus application and the cropping pattern on the maize grain yield. However, AE, RE, and PFP showed significant effects (*p* < 0.05) (Figure 4, 1st year). Conversely, the third year (*p* < 0.05) and seventh year (*p* < 0.01) intervals revealed that cropping patterns and phosphorus application significantly directly and indirectly affected maize grain yield, SYI, AE, RE, and the PFP of phosphate fertilizer.

Additionally, the direct impacts of phosphorus application on AE, RE, PFP, and SYI were positively correlated. The direct impact of phosphorus fertilization and the cropping pattern on the maize grain yield, AE, RE, PFP, and SYI exceeded the indirect effect (Figure 4, 3rd and 7th year intervals). The intercropping pattern exhibited significant direct and indirect positive effects on the maize grain yield, AE, RE, PFP, and sustainability index year intervals. However, it negatively affected SI (Figure 4, 3rd and 7th year intervals). Furthermore, within the third year interval, the cropping pattern, phosphorus application, AE, RE, and PFP had positive effects on the yield (*p* < 0.05) (Figure 4, 3rd year interval). Similarly, over the seventh year interval, AE, RE, PFP, and SYI had a substantially more significant direct positive impact (*p* < 0.01) (Figure 4, 7th year interval). Under intercropping, phosphorus fertilizer affects maize grain production productivity and sustainability (Figure 4).

## 3. Discussion

### 3.1. Maize Yield and Yield Sustainability Affected by Year Intervals Under Continuous Intercropping

Cereal–legume intercropping is used worldwide to boost the crop yield, conserve natural resources, enhance fertilizer use efficiency, and create a sustainable and environmentally friendly agricultural system [40,41,42]. Maize and soybean intercropping is a prevalent cultivation practice in China [43]. This is because maize and soybean, the leading food and feed grains, are essential for food production in many nations. Meanwhile, certain analysts propose that China’s food security is more dependent on animal feed grain security, mainly maize and soybeans [44]. Accordingly, we investigated year intervals of continuous intercropping trials for maize and soybean. Specifically, we studied the intercropped maize grain yield in the third (2019) and seventh (2023) year intervals and the first year (2017). Our study results confirmed that maize/soybean continuous intercropping outperformed monocropping consistently across the trial years, demonstrating its effectiveness in increasing maize productivity [14,45]. In line with [14,15,37], in the third and seventh year intervals (2019, 2023), our intercropped maize increased the yield by 35% and 58% above the maize monoculture yield, respectively. Numerous studies have shown that a grain yield advantage depends on resource optimization and the intraspecific/interspecific interaction balance [46,47,48]. Complementarity interactions result in a grain yield advantage when intercrops receive the same or different resources at different times or places. This lowers niche overlap and competition and facilitates intercrop relations [16,49,50,51,52]. Moreover, we observed highly significant effects of the year (Y), cropping pattern (C), phosphorus application (P), and their interaction on maize grain yield (Table 1). These findings show that maize intercropping can offer a sustainable way to boost the maize yield in varied agroecological situations and changing environmental circumstances.

Our year-interval studies indicated that intercropping and phosphorus application increase sustainability (SYI) and maize production. The SEM showed that intercropping and phosphate application enhanced maize yield sustainability. The third and seventh year intervals of sustainability of maize intercropping yields exhibited a significant increase (*p* < 0.01) compared to monocultures when the same amount of *p* was applied (Figure 1b). Increased SYI values were recorded at 23% in the initial year of 2017, 25% in the third year of 2019, and 35% in the seventh year of 2023, which aligns with the findings of [14,45,53]. Our experimental trials showed a progressive increase from the beginning of the experiment. The SYI reached its peak value in the seventh year interval of 2023. This indicates that maize intercropping maintains and sustains the production of grain yields over time. Our findings indicated that the intercropping yield sustainability was consistently better than that for monoculture (Figure 1b). The outcomes are consistent with the findings of Liu, Meng [45], where maize/soybean intercropping was demonstrated to have a superior yield sustainability compared to monoculture.

### 3.2. Actual Yield Loss and Yield Stability Affected by Year Intervals Under Continuous Intercropping

The term yield per plant-derived AYL index measures competition across and within crops more precisely and explains each intercropping species’ behavior well. In contrast, the partial AYL symbol and numerical value show yield variations, either gains or losses [54,55]. In this study, the year intervals and the cropping pattern significantly impacted AYL values (*p* < 0.05), showing that the effects of intercropping on maize yield varied under the conventional phosphorus application over the years. This pattern aligns with previous studies [45,52,56,57], which also found that maize holds a competitive edge in maize/soybean intercropping systems. Consistent with Zhou, Su [15], we observed a significant effect (*p* < 0.05) of the cropping pattern, phosphorus application, and their interaction with AYL values (Figure 2a). The structural equation modelling (SEM) analysis showed that the cropping pattern and conventional P application directly impacted the maize yield in third and seventh year intervals. These findings align with the results obtained from the AYL analysis.

Intercropping improves crop yield stability by employing the concept of compensation, which decreases the probability of both crops being simultaneously lost due to causes such as pests, disease, or severe weather [40]. In 1980, ref. [58] conducted 94 experiments to test this idea. Other intercropping species might use the resources to maintain their yield whenever an intercropping species is infected. Resource niche complementarity is vital to yield stability. After four positioning tests on soils with different fertility levels, Li found intercropping had better temporal stability than monoculture. Intercropping increases total nitrogen, soil organic matter, and macro-aggregates, which improves soil fertility and yield stability [14]. Multiple studies have demonstrated that intercropping crop diversification increases system production and stability [59,60,61]. A meta-analysis of 33 research studies published in 1980 [40] supported the findings of our study. Our year-interval studies indicated that continuous intercropping and phosphorus fertilizer application diminish variability, stabilizing maize production. The SEM analysis revealed that phosphate fertilization and the intercropping system enhanced maize yield sustainability. The year intervals for the sustainability of maize intercropping yields exhibited a significant increase (*p* < 0.01) compared to monocultures (Figure 5, 7th year). This outcome aligns with the findings of [45], where maize/soybean intercropping demonstrated superior yield stability compared to monoculture.

### 3.3. Phosphorus Fertilizer Contribution and Phosphorus Use Efficiency Affected by Year Intervals of Continuous Intercropping

The intercropping system of maize and soybean demonstrated a significant advantage regarding phosphorus utilization efficiency. The fertilizer contribution refers to the impact of fertilization on crop output, indicating the annual productivity of fertilizer input [62]. The current study found that the fertilizer contribution to the yield increased across the trial years. Under maize/soybean intercropping, the phosphorus fertilizer contribution rose with planting years; this implies that fertilizer and complementary *p* supplies increased the maize intercropping system’s *p* consumption. Consistent with the findings of Liu, Meng [45] and Zhou, Su [15], compared to monoculture, intercropping maize enhances the phosphorus fertilizer contribution by 10% to 16% across the year intervals, consequently increasing phosphorus use efficiency. Thus, enhancing the soil’s inherent fertility can decrease the reliance on external fertilizers for maize production, reducing the quantity of fertilizers needed.

The agronomic efficiency (AE) and the apparent recovery efficiency (RE) of applied phosphate fertilizer indicate the phosphorus use efficiency [63,64]. In this study, AE and RE were positively correlated with the year and cropping patterns (*p* < 0.01) (Figure 3b,c). In line with the prior studies of An, Yu [65] and Darch, Giles [66], the current trial results indicated that intercropping enhanced the (AE), (RE) at P90 kg ha^−1^, suggesting that maize/soybean mixed cropping resulted in overyielding in terms of the phosphorus content per unit of phosphate fertilizer application. Consequently, it revealed the ability of intercropping to promote agricultural sustainability by saving on phosphorus fertilizer.

Similarly, our experimental results revealed that the maize intercropping system at the conventional phosphorus fertilizer application rate increased the partial factor productivity of phosphorus fertilizer (PFP) at a highly significant level (*p* < 0.05) (Figure 3d). Under intercropping systems, there was an increased yield, which could be ascribed to an increase in resource exploitation. For sustainability, under intercropping systems, there is enhanced resource use, resulting in increased *p* use efficiency in intercropping. It is expected that with time, the available soil resources will mineralize to release the bound nutrient, making it available for plant uptake. This could significantly impact both the maize yield and phosphorus use efficiency. Yang, Zhang’s [36] research results showed that the maize/soybean intercropping system greatly affects the partial factor productivity of phosphorus fertilizer, which further supports the results of our study. After examining our results, we can conclude that the maize grain yield and phosphorus use efficiency (AE, RE, and PFP) of the maize/soybean intercropping system are enhanced mainly because of the facilitative and complementary interaction between soybean and maize, which represents an interplanting advantage [36].

## 4. Materials and Methods

### 4.1. Experimental Design and Agronomic Management

Consecutive experimental trials were established between 2017 and 2023 on low-fertile, acidic soil at the Xiaoshao Research Center, Kunming Yunnan province, China (24°54′ N, 102°41′ E). The region exhibits a northern monsoon subtropical climate characterized by an average year-round temperature of 14.4 °C and an annual rainfall level of 850 mm. The annual precipitation and mean temperature in the maize season during 2017, 2019 and 2023 (Figure 5). The red soil of the soil plateau is classified as Ferralsols in the USDA classification. This particular soil exhibits a high phosphorus fixing capacity. Before the start of this study, in 2017, the basic physicochemical characteristics of the top 20 cm of soil were as follows: bulk density 1.36 mg·cm^−3^, pH 4.53 (water-to-soil ratio of 2.5:1), organic matter content 4.50 g·kg^−1^, nitrate–nitrogen concentration 2.19 mg·kg^−1^, total phosphorus (TP) 0.19 g kg^−1^, and Olsen phosphorus 4.02 mg·kg^−1^.

The field experiment was conducted using a randomized complete block design in 2017 to test two cropping patterns: maize monocropping (MM) and maize intercropping with soybean. Six plots (4 m × 6 m = 24 m^2^) were employed, each with two treatment combinations and three replications. According to our study objective, we selected and analyzed the data at specific year intervals. The first year of the experiment, 2017, was selected as the first interval (expressed as 1st year). Similarly, the third year of the experiment, 2019, was considered the third year interval (expressed as 3rd year interval), and the seventh year of the experiment, 2023, was considered the 7th year interval. Each maize and soybean crop was planted in a six-row intercropping arrangement, while twelve rows of maize were cultivated in monoculture. The crop spacing was two rows of soybeans and two rows of maize. The spacing between rows of maize plants was 50 cm, and the gap between plants was 25 cm, resulting in 75,000 plants·ha^−1^. The row spacing was 50 cm, the edge distances were 25 cm, and the intercropping between the maize and soybean was 25 cm. The Yunrui 88 maize variety and the Kaiyu-2 soybean cultivar were grown from May to October or November between 2017 and 2023. For fertilizer management, urea, superphosphate, and potassium sulfate fertilizers were applied for nitrogen (N), phosphorus (P), and potassium (K). We applied 100 kg N ha^−1^, 90 kg P_2_O_5_ ha^−1^, and 75 kg K_2_O ha^−1^ each year before cultivating maize and soybean crops. At different maize growth stages (V6, V12), an extra 62.5 and 87.5 kg of nitrogen were added as topdressing to monoculture and intercropping plots. The land management practices were similar across all treatments of maize and soybean.

### 4.2. Sample Collection

At full maturity, the middle rows of every experimental block of maize, including monocropping and intercropping, were harvested manually, and the data were collected each year. The sampled plants were air-dried until they attained a consistent weight. Each air-dried maize plant was manually removed from its husk and measured for dry matter and grain yield. Sample yields were in kg·ha^−1^.

### 4.3. Yield Indices Calculation

#### 4.3.1. Maize Grain Yield and Sustainable Yield Index

The term delta “Δ” is a relative quantity symbol that is used to measure the differential change in crop yield of intercropping and monocropping systems. We calculated (ΔYield), (ΔSYI), (ΔSI), (ΔCRPF), (ΔAE), (ΔRE), and (ΔPFP) reported by Zhou, Su [15] to evaluate the changes in maize grain yield and sustainability over different year intervals. These measurements were performed for the seven-year average, first year (2017), third year (2019), and seventh year (2023) data, following [45] and [67] (Equations (1) and (2)).Yield (kg ha^−1^) = average grain yield per plant (kg) × planting density (plant·ha^−1^) × 10^−6^(1)

The Sustainable Yield Index (SYI) was calculated, offering an index to measure yield sustainability.(2)$$SYI=\frac{\bar{Y}-\sigma }{Ymax}\;$$

$where\;\bar{Y}$ is the average yield for each treatment during the experimental period, and “σ” is the standard deviation of each treatment’s yield during that time. Ymax represents the maximum yield attained throughout the experimental period for each treatment. SYI is between 0 and 1. A greater SYI denotes a more sustainable production.

#### 4.3.2. Actual Yield Loss and Maize Yield Stability

The actual yield loss (AYL) index is based on per-plant yield and is used to give more accurate information about the component crops and the behavior of each species in the intercropping system. The AYL is the intercrop proportionate yield loss or gain relative to the monocrop. It can have negative and positive values, demonstrating the per-plant disadvantage or advantage of the intercropping system [68]. The actual yield loss was computed as indicated in Equation (3), reported by [54] in 2018.(3)$$\mathrm{Actual}\;\mathrm{yield}\;\mathrm{loss}=\mathrm{AYLm}+\mathrm{AYLs}=\left(\frac{\frac{Yim}{Pim}}{\frac{Ym}{Pm}}\right)-1+\left(\frac{\frac{Yis}{Pis}}{\frac{Ys}{Ps}}\right)-1$$

Pm and Pim are monoculture and intercropping maize proportions. Both proportions are 1. When the actual yield loss value exceeds 0, intercropped crop x outperforms the monoculture in yield. The intercropped crop × yield will be inferior to the monoculture yield if AYLx < 0. Consequently, intercropping offers no benefits.

The coefficient of variation (CV) is commonly used in plant ecology to assess the yield stability index by measuring the yield variation degree between years. The lower the value, the higher the yield stability. The stability index was calculated as follows in Equation (4) and reported by [67]:(4)$$CV=\frac{\;\sigma }{\bar{Y}}$$
where $\bar{Y}$ is the average yield per treatment and σ is the standard deviation of each treatment yield.

#### 4.3.3. The Contributions of P Fertilizer (CPF), Agronomic Efficiency (AE), the Apparent Recovery Rate of Phosphate Fertilizer (RE), and the Partial Productivity of Phosphate Fertilizer (PFPP)

Soil fertilizer contribution rates were evaluated using the formulas provided in Equation (5) and reported by [69,70]:(5)$$Contribution\;of\;fertilizer\;\left(\%\right)=CGYF-\frac{CGYUF}{CGYF}\times 100\;$$CGYUF = crop grain yield in the unfertilized treatmentMCGYF = maximum crop grain yield in the fertilized treatment

The agronomic efficiency of phosphorus fertilizer was measured as follows in Equation (6) and reported by [63].(6)AE=Yf−Y0/F
where AE is the agronomic efficiency of phosphate fertilizer, kg·kg^−1^; Yf is the economic yield of crops in the phosphorus application plot, kg·hm^−2^; Y0 is the economic yield of crops in the non-phosphorus area, kg·hm^−2^; and F is the amount of phosphate fertilizer input, kg·hm^−2^.

The apparent recovery efficiency of phosphate fertilizer was measured in Equation (7), reported by Xia, Zhao [71].(7)RE=U1−U0/F×100
where RE is the apparent recovery efficiency of phosphate fertilizer, %; U1 and U0 are the accumulation of phosphorus nutrients in crops in the phosphorus application area and the non-phosphorus application area, respectively, kg·hm^−2^; and F is the amount of phosphate fertilizer, kg·hm^−2^.

The PFP of phosphorus application was measured as follows in Equation (8), reported by [63].(8)PFPP kg.kg−1=Y/F
where PFP is the partial productivity of phosphate fertilizer, kg·kg^−1^; Y is the yield of phosphate fertilizer crops, kg·hm^−2^; and F is the amount of phosphorus fertilizer, kg·hm^−2^.

### 4.4. Statistical Analysis

Statistical analysis was performed using the Statistix 8.1 software (Analytical Software, Tallahassee, FL, USA) Tuckey’s HSD multiple-comparison method was applied to test the significance of differences (α = 0.05). For the structural equation model (SEM), the lavaan package was used in R. This model was applied to check the relationships among the yield, phosphorus application, yield sustainability, and phosphorus use efficiency indices (AE, RE, and PFP) under the intercropping system for year intervals. The model’s adequacy was assessed using *p*-values (*p* > 0.05), the root means square residual (RMR < 0.05), the goodness of fit (GFI > 0.9), and the comparative fit index (CFI > 0.9) [72].

## 5. Conclusions

Our study reveals that continuous maize/soybean intercropping significantly enhances the crop yield and phosphorus use efficiency, particularly in resource-constrained environments. When averaged over seven years, the maize yield increased by 58%, while the agronomic efficiency, recovery efficiency, partial factor productivity, and crop phosphorus uptake efficiency improved by 60%, 58%, 24%, and 10.5%, respectively. Moreover, the pronounced benefits observed in the seventh year indicated that prolonged intercropping fosters cumulative improvements in soil fertility, nutrient cycling, and system resilience. Hence, the intercropping system facilitated better nutrient cycling, reduced nutrient competition, and enhanced system sustainability, leading to a higher phosphorus use efficiency across the different year intervals. Overall, these findings highlight the long-term advantages of intercropping in optimizing resource use efficiency and stabilizing yields, making it a viable strategy for sustainable agricultural intensification, especially in the context of climate change and limited resources. Therefore, given the growing challenges of food security and soil degradation, the adoption of long-term intercropping systems could play a crucial role in enhancing productivity and nutrient management in resource-limited environments.

## Figures and Tables

**Figure 1 plants-14-01060-f001:**
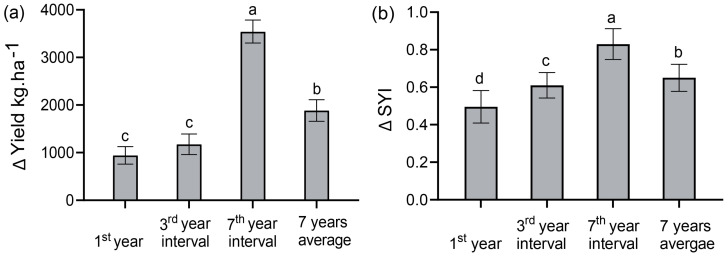
Effect of year intervals of maize/soybean continuous intercropping on the (**a**) change in maize grain yield and (**b**) yield sustainability (SYI) in the 1st year, the 3rd year interval in 2019, the 7th year interval in 2023, and an average of 7 years. Bars denote the average of 3 replications ± SD, while the standard deviation is denoted by error bars. Different lowercase letters indicate differences between the cropping patterns under the same phosphate fertilization level with an LSD level of 5%. Note: The 1st year is 2017; the 3rd year is the 3rd year interval in 2019; the 7th year is the 7th year interval in 2023.

**Figure 2 plants-14-01060-f002:**
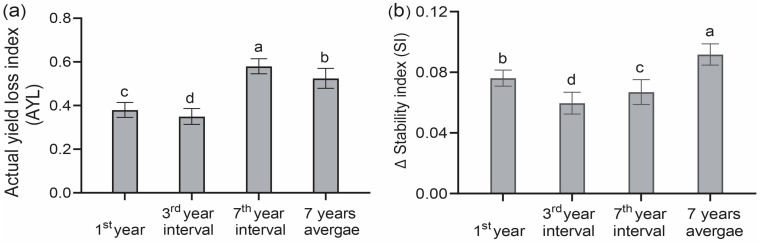
Effect of year intervals of maize/soybean continuous intercropping on (**a**) actual yield loss (AYL) and (**b**) stability index (SI) in the 1st year, the 3rd year interval in 2019, the 7th year interval in 2023, and an average of 7 years. Bars denote the average of 3 replications ± SD, while the standard deviation is denoted by error bars. Different lowercase letters indicate differences between the cropping patterns under the same phosphate fertilization level with an LSD level of 5%. Note: The 1st year is 2017; the 3rd year is the 3rd year interval in 2019; the 7th year is the 7th year interval in 2023.

**Figure 3 plants-14-01060-f003:**
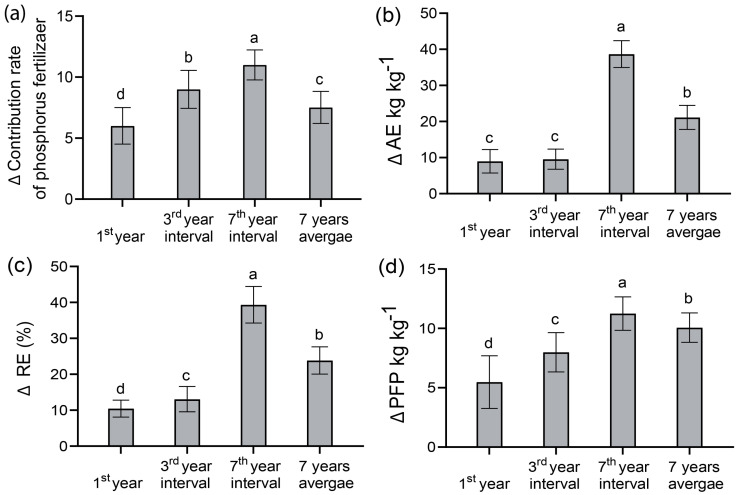
Effect of year intervals of maize/soybean continuous intercropping on the (**a**) contribution rate, (**b**) agronomic efficiency (AE), (**c**) relative efficiency (RE), and (**d**) partial factor productivity (PFP) of phosphate fertilizer in the 1st year, the 3rd year interval in 2019, the 7th year interval in 2023, and an average of 7 years. Bars denote the average of 3 replications ± SD, while the standard deviation is denoted by error bars. Different lowercase letters indicate differences between the cropping patterns under the same phosphate fertilization level with an LSD level of 5%. Note: The 1st year is 2017; the 3rd year is the 3rd year interval in 2019; the 7th year is the 7th year interval in 2023.

**Figure 4 plants-14-01060-f004:**
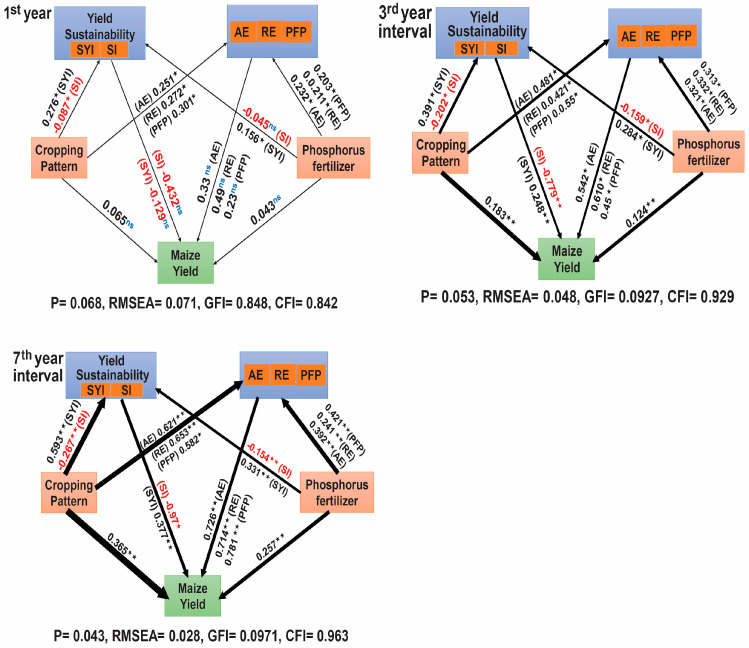
SEM effect of the cropping pattern, phosphorus application, SYI, SI, AE, RE, and PFP on maize yield in the 1st year, the 3rd year interval in 2019, and the 7th year interval in 2023. Note: The 1st year is 2017; the 3rd year is the 3rd year interval in 2019; the 7th year is the 7th year interval in 2023. For each predictor, the correlation coefficients are represented by the numbers above and below the arrows, and the arrows’ widths represent the correlations’ strengths. Significance is indicated by * and ** at *p* <0.05 and <0.01, respectively; ns represents no significance (*p* > 0.05).

**Figure 5 plants-14-01060-f005:**
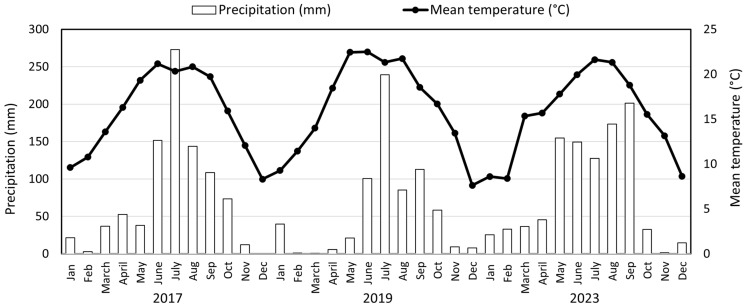
Annual precipitation and mean temperature in the maize growing season during 2017, 2019, and 2023.

**Table 1 plants-14-01060-t001:** Three-way ANOVA of change in maize grain yield from 2017 to 2023 under continuous maize/soybean intercropping and maize monoculture systems at a conventional phosphorus rate.

Source of Variation	Maize Yield	
	Df	F
Year (Y)	2	99.01 ***
Phosphorus level (P)	1	1032.20 ***
Cropping pattern (C)	1	66.26 ***
Year × Phosphorus level (Y × P)	2	184.94 ***
Year × cropping pattern (Y × C)	2	9.58 ***
Phosphorus level × Cropping pattern (P × C)	1	46.07 ***
Year × Phosphorus level × Cropping pattern (Y × P × C)	2	12.23 ***

Note: Degrees of freedom denoted by DF; F value denoted by F; “***” represents *p* < 0.001.

## Data Availability

The original contributions presented in this study are included in the article; further inquiries can be directed to the corresponding author.

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
