# Peer review of "Long-Term Year-Interval Effect of Continuous Maize/Soybean Intercropping on Maize Yield and Phosphorus Use Efficiency"

_plants, 2025, doi:10.3390/plants14071060_

Round 1
Reviewer 1 Report
Comments and Suggestions for Authors
I think that the design of the experiment could include also an alternation of crops (corn and soybean) in time and space,because very often alternation of soybean and corn is typical for some regions of the world.
It would be desirable to have also the continuous(monocropping) corn on fertilized and unfertilized plots,which would allow to determine the P-use efficiency in time for intercropping of maize and soybeans as well as for monoculture.It is evident that the yields of corn in monoculture will decrease but probably the extra yield of corn from fertilizers could increase as well as nitrogen -use efficiency.
This could emphasize the crucial role of soil health,the health of the root system.
The article doesn"t have an answer and explanation for the increase P-use efficiency in intercropping ,especially in seven years interval.
It would be nice to determine the influence of michorhiza in permanent cropping and intercropping.
Interesting to notice that in the first year of the experiment(2017)no direct impact of phosphorus application and cropping patern on maize grain yield were found.Cropping pattern and P-fertilizers affected yields of corn on the 3 and 7 years. Intercropped maize yield by longer period of time with soybeans outperformed monocropping. Why to use monocropping of corn for grain?
In seven years interval maize yield increase with 55% on average as well as P-use efficiency increased ,but what could happen under the influence of organic fertilizers?What has happened during 7 years under continuous corn for grain and under intercropping of maize with soybean?It is evident that such a crop system isn't sustainable only under the influence of mineral fertilizers.The authors don't show the level of infestation with weeds,pests and deseases.
Yields are higher by intercropping if to compare with continuous corn,but how efficient are suplementary expenditures for P application?How they are recovering?
What indexes should be used for determining the inherent soil fertility in order to decrease reliance on external fertilizers?May be is better to avoid permanent cropping of corn for grain.
I would like to encourage authors to continue their research but in the same time to try to answer some of the questions which have been raised here.
Author Response
Thank you for your letter and the reviewer’s comments concerning our manuscript entitled “Effects of Year-intervals of Continuous intercropping on maize 2 Yield and Phosphorus use Efficiency modified as Long-Term Year-interval effect of Continuous Maize/soybean intercropping on Maize Yield and Phosphorus use Efficiency” (ID: plants-3471797). We are very grateful to you and the reviewer’s critical comments and thoughtful suggestions. We have carefully studied the comments and suggestions and have made modifications, which we hope meet with approval. Revised portions are marked in red in the revised manuscript. Some of your questions were answered below.
Once again, we acknowledge your and the reviewer’s comments and constructive suggestions very much, which are valuable in improving the quality of our manuscript.
Kind regards
Sincerely
Reviewer 1:
Comments 1: I think that the design of the experiment could also include an alternation of crops (corn and soybean) in time and space because, very often, alternation of soybean and corn is typical for some regions of the world.
Response:
Thank you for your suggestion. We agree that crop rotation, mainly alternating corn and soybean, is a common practice that can positively affect soil health, pest management, and yields. However, our study specifically focused on the long-term effects of continuous maize//soybean intercropping on maize yield, phosphorus uses efficiency (PUE), and sustainability over year intervals. While our research did not explore crop rotation, we recognize that alternating crops could offer additional insights, particularly regarding soil health and resource use efficiency. In future research, we plan to explore this aspect to understand its impact on soil quality, yield stability, and nutrient management.
Comment 2:
It would be desirable to have continuous (monocropping) corn on fertilized and unfertilized plots, which would allow us to determine the P-use efficiency in time for the intercropping of maize and soybeans and for monoculture. It is evident that the yields of corn in monoculture will decrease, but the extra yield of corn from fertilizers could probably increase, as well as nitrogen-use efficiency. This could emphasize the crucial role of soil health, the health of the root system.
Response 2:
Thank you for your valuable suggestion. We agree that including monocropping maize with both fertilized and unfertilized plots would provide additional insights into phosphorus (P) and nitrogen-use efficiency. However, in our study, we have calculated the delta yield, which represents the difference between the maize yield in intercropping and monocropping systems. This approach allowed us to focus on the yield differential between these two systems, effectively incorporating the monocropping data into the analysis. Therefore, we did not separately present the monocropping yield data. In terms of fertilized and unfertilized, we do have the unfertilized plot data. We apologize for any confusion regarding the unfertilized treatment. Let me clear that we did include an unfertilized treatment in our experiment, our main focus in the manuscript was on the changes in maize yield associated with intercropping compared to monocropping over year intervals. Therefore, we did not explicitly discuss the unfertilized treatment in the main text, as our primary objective was to highlight the yield differences between intercropping and monocropping. To ensure transparency, we will provide the unfertilized plot data in the as below in Figure A.
Supplementary Fig. A. Maize grain yield in fertilized and unfertilized plots under maize//soybean continuous intercropping in
Comments 3: The article doesn"t have an answer and explanation for the increased P-use efficiency in intercropping, especially in seven year intervals
Response 3: The increased phosphorus-use efficiency (PUE) in maize-soybean intercropping, particularly in the seventh-year Interval, can be attributed to several interrelated factors over time as shown in line 338- 344:
- Complementary Root Systems: Maize and soybean have contrasting root architectures, with maize having deeper roots and soybean being more efficient at mobilizing phosphorus. Over time, this complementary root interaction enhances P uptake efficiency.
- Rhizosphere Interactions: Soybean, as a legume, acidifies the rhizosphere, releasing organic acids that solubilize phosphorus, making it more available for both crops. This effect likely becomes more pronounced as the system matures.
- Soil Organic Matter and Microbial Activity: Continuous intercropping over several years enhances soil organic matter and microbial activity, which play a key role in mineralizing organic phosphorus into plant-available forms, improving P uptake efficiency.
- Reduced Competition and Improved Resource Partitioning: Over time, maize and soybean adapt to each other’s presence, leading to reduced competition and better resource partitioning, which enhances phosphorus utilization, especially in the later years of the intercropping system.
Comments 4: It would be nice to determine the influence of mycorrhiza in permanent cropping and intercropping.
Response: 4
Thank you for your valuable suggestion regarding the inclusion of the mycorrhizal effect in permanent cropping and intercropping systems. While we acknowledge that mycorrhiza can play an important role in nutrient uptake, particularly in phosphorus absorption, we did not specifically investigate the mycorrhizal effect in this study. This is similar to the approach taken by other recent studies, such as Zhou et al. (2024) and Liu et al. (2023), who also focused on maize-soybean intercropping to assess maize yield and stability under different fertilizer levels but did not incorporate the mycorrhizal effect in their analysis. Despite this, both of these studies have been successfully published. Given the focus of our study on phosphorus use efficiency, cropping patterns, and maize yield dynamics over time, the inclusion of mycorrhizal analysis was beyond the scope of the current research. However, we appreciate the importance of mycorrhizal interactions and will consider it for future studies. References:
Liu et al. (2023). Maize/soybean intercrop over time has higher yield stability relative to matched monoculture under different nitrogen-application rates. Field Crops Research, 301, 109015.
Zhou et al. (2024). Maize//Soybean Intercropping Improves Yield Stability and Sustainability in Red Soil under Different Phosphate Application Rates in Southwest China
Comments 5: It is interesting to notice that in the first year of the experiment (2017), no direct impact of phosphorus application and cropping pattern on maize grain yield was found. Cropping pattern and P-fertilizers affected yields of corn in the 3 and 7 years. Intercropped maize yield over an extended period of time with soybeans outperformed monocropping. Why use monocropping of corn for grain?
Response 5:
Thank you for your valuable suggestion. As observed in the SEM analysis, the lack of a direct impact of phosphorus application and cropping pattern on maize grain yield in the first year (2017) may be attributed to the limitations of SEM with small sample sizes, which require larger datasets to detect robust relationships. Additionally, in the first year, competition for resources between maize and soybean could have increased, which restricted both crops' growth and yield in the intercropping system, resulting in lower system yield. As the crops adapted over time, resource competition decreased, and complementarity maximized, allowing intercropping to outperform monocropping in later years. In response to the use of corn monocropping, maize monocropping served as a baseline to compare the benefits of intercropping with soybean. By including maize monocropping, we could better assess the specific contributions of intercropping to yield enhancement, phosphorus use efficiency, and sustainability over time.
Comments 6: In seven years, interval, maize yield increased by 55% on average, and P-use efficiency increased, but what could happen under the influence of organic fertilizers? What has happened during the 7 years under continuous corn for grain and the intercropping of maize with soybeans? Such a crop system isn't sustainable only under the influence of mineral fertilizers. The authors don't show the level of infestation with weeds, pests and diseases.
Response 6:
We agree that the sustainability of crop systems cannot rely solely on mineral fertilizers, and including organic fertilizers could significantly impact both maize yield and phosphorus use efficiency. Our study focused on the effects of different interval gaps on P-use efficiency in maize-soybean intercropping. It did not evaluate the influence of organic fertilizers, which would be an interesting avenue for future research. Over the seven-year interval, the maize yield and PUE improvement in the intercropping system can be attributed to the crops' gradual adaptation and the intercropping system's complementary benefits. However, we acknowledge that pest, weed, and disease management are also crucial factors for long-term sustainability, and the effects of these factors were not addressed in our study, as our study has just focused on yield and nutrient efficiency on intercropping. We appreciate the suggestion and will consider including a more comprehensive discussion on these aspects and the potential role of organic fertilizers in future work.
Comments 7: Yields are higher by intercropping if compared with continuous corn, but how efficient are supplementary expenditures for P application? How are they recovering?
Response 7:
While intercropping resulted in higher yields than continuous maize, supplementary P application's economic efficiency depends on factors such as soil P availability, crop uptake, and yield gains over time. The higher P-use efficiency observed in intercropping suggests better nutrient recovery, but a detailed cost-benefit analysis would be needed to assess the economic return on P investment fully. Future research could explore this aspect in more detail.
Comments 8: What indexes should be used for determining the inherent soil fertility to decrease reliance on external fertilizers? Maybe it is better to avoid permanent cropping of corn for grain.
Response 8: Thank you for your comment. Soil fertility can be assessed using organic matter, available P, CEC, microbial activity, and the interspecific facilitation effect of cereal-legume intercrops. It is well established that legume-cereal intercropping enhances the availability of both phosphorus and nitrogen, improving nutrient cycling and reducing the need for external fertilizers. This facilitative interaction between maize and soybean promotes a more sustainable system by improving soil fertility naturally over time. In our study, the increased phosphorus uses efficiency and overall yield in intercropping further support the benefits of interspecific facilitation in reducing reliance on mineral fertilizers. Thus, opting for continuous intercropping could improve nutrient cycling and reduce fertilizer dependence.
Comments 9: I would like to encourage authors to continue their research but in the same time to try to answer some of the questions which have been raised here.
Thank you for your encouragement and thoughtful feedback. We have carefully addressed the questions raised in your review, making revisions where necessary and providing clarifications in our response file. We believe the changes and explanations strengthen our manuscript, and we are committed to continuing this line of research. We appreciate your constructive comments and will certainly consider them in future work to further advance our understanding of the topic.

Reviewer 2 Report
Comments and Suggestions for Authors
Manuscript entitled “Effects of Year-intervals of Continuous intercropping on maize Yield and Phosphorus use Efficiency”. The manuscript investigated the effect of continuous maize//soybean intercropping on maize yield, yield sustainability and PUE. Minor points need to be addressed before accepted.
- The manuscript set maize monocropping as control, and investigate the effect to maize yield and yield sustainability. Why not set soybean monocropping as control?
- Line 22 and 26. Full names of AE, RE, PFP, CPF and SEM are needed.
- Line 384 and 402. Revised 2.3.2 to 4.3.2 and 2.3.3 to 4.3.3.
Author Response
Response to Reviewers 2 Comments
Answer to the comments:
Thank you for your letter and the reviewer’s comments concerning our manuscript entitled “Effects of Year-intervals of Continuous intercropping on maize 2 Yield and Phosphorus use Efficiency modified as Long-Term Year-interval effect of Continuous Maize/soybean intercropping on Maize Yield and Phosphorus use Efficiency” (ID: plants-3471797). We are very grateful to you and the reviewer’s critical comments and thoughtful suggestions. We have carefully studied the comments and suggestions and have made modifications, which we hope meet with approval. Revised portions are marked in red in the revised manuscript. Some of your questions were answered below.
Once again, we acknowledge your and the reviewer’s comments and constructive suggestions very much, which are valuable in improving the quality of our manuscript.
Kind regards
Sincerely
Reviewer 2: Comments 1: The manuscript set maize monocropping as control and investigated the effect on maize yield and yield sustainability. Why not set soybean monocropping as control?
Response 1:
Thank you for your comment. In our study, we set maize monocropping as the control because the primary focus was to evaluate the effects of intercropping on maize yield and phosphorus use efficiency (PUE). Since maize is the main crop of interest, using maize monocropping as the control allows us to assess the benefits of intercropping maize with soybean. The omission of a soybean monocropping treatment was intentional, as our primary focus was to evaluate the impacts of continuous maize//soybean intercropping on maize yield, phosphorus use efficiency, and sustainability. We aimed to compare maize monocropping with maize//soybean intercropping to highlight the advantages of intercropping, particularly in resource use efficiency and crop yield stability. Including a soybean monocropping treatment was not necessary for the scope of our study. However, we recognize that soybean monocropping could provide valuable insights into soybean’s individual performance and phosphorus use efficiency, and this could be explored in future studies.

Reviewer 3 Report
Comments and Suggestions for Authors
The manuscript investigated the yield and variation of maize and soybean under monocropping and intercropping in the subtropical region. Lots of such researches have been taken, while here two points benefit its possible for consideration for possible publication in Plants, one is that the experiment lasts seven years, and the second was that it considers phosphorous effects and efficiency, while only odd year results were presented. While it had lots of key flaws needed to be eliminated for further consideration, and the two key complains were experimental design and data calculation, and presentation. Some technologies should be unified, e.g. year-interval and year interval (actually this term was not clearly explained in the text), maize/soybean intercropping, maize//soybean intercropping, etc. Besides these, some were as following:
- Title: because maize and soybean was annual crop, thus planting year was not the key factor affecting its yield performance, while rainfall and distribution, soil fertility content, row spacing, crop variety, thus the title was not accurate. if P was an experiment factor, which should appear in the title also.
- Introduction: It is a feature of this study to discuss the effect of P fertilizer, except in the subtropical area, but the introduction part rarely introduces the research progress under P fertilizer intercropping, some more information needed. Line73-74: not clear.
- Materials and Methods: there were too many complains here. why no soybean monocropping design? why only analyze the 1, 3 and 7year, but no 2, 4 and 6? odd year, not even year? the plot size was 4*6, and how the maize and soybean was arranged under such row spacing and plant spacing? Line 360:not clear, same amount N for monocropping and intercropping? the yield was air-dried for maize and soybean? Line 368:the illustration here was of some redundant, also the equation (1). Line 384:all the calculated index need further checking, for the explanations and significance, and also calculation method. Line 400: the explanations was not clear and correct, here no repeat for monoculture or mixture, only year difference. Line 407:it seemed no unfertilized treatment here, and how the data came from? Line 430: the statistical analysis illustration was not clear, and SEM method not correct as shown in figure 4 because the index calculation was based on yield.
- Conclusion: not fully and comprehensively summarized, which should be mixture was appropriate for maize and soybean in stabilizing yield and nutrient use efficiency.
- Others: the yield of the two crops under mono- and mixture for each experimental year and the rainfall during the growth seasons for the seven years should be the results for the manuscript. Some were marked and annotated in the manuscript.

the terminology should be unified, and the illustration logicality need improvmet.
Author Response
Response to Reviewers 3 Comments
Answer to the comments:
Thank you for your letter and the reviewer’s comments concerning our manuscript entitled “Effects of Year-intervals of Continuous intercropping on maize 2 Yield and Phosphorus use Efficiency modified as Long-Term Year-interval effect of Continuous Maize/soybean intercropping on Maize Yield and Phosphorus use Efficiency” (ID: plants-3471797). We are very grateful to you and the reviewer’s critical comments and thoughtful suggestions. We have carefully studied the comments and suggestions and have made modifications, which we hope meet with approval. Revised portions are marked in red in the revised manuscript. Some of your questions were answered below.
Once again, we acknowledge your and the reviewer’s comments and constructive suggestions very much, which are valuable in improving the quality of our manuscript.
Kind regards
Sincerely
Comments and Suggestions for Authors
The manuscript investigated the yield and variation of maize and soybean under monocropping and intercropping in the subtropical region. Lots of such researches have been taken, while here two points benefit its possible for consideration for possible publication in Plants, one is that the experiment lasts seven years, and the second was that it considers phosphorous effects and efficiency, while only odd year results were presented. While it had lots of key flaws needed to be eliminated for further consideration, and the two key complains were experimental design and data calculation, and presentation. Some technologies should be unified, e.g. year-interval and year interval (actually this term was not clearly explained in the text), maize/soybean intercropping, maize//soybean intercropping, etc. Besides these, some were as follows:
Response:
Thank you for your thoughtful feedback. We appreciate the recognition of the long-term nature of our study and the focus on phosphorus efficiency, which distinguishes our research. We agree that some terms need further clarification, and we have unified terminology such as "year-interval" and "maize/soybean intercropping" in the revised manuscript to ensure consistency and clarity. In this study, the term year-interval refers to the specific time points selected to assess the effects of maize//soybean intercropping on maize yield and phosphorus use efficiency. Given the seven-year duration of the field experiment, data were collected at three key intervals: the first year, the third year, and the seventh year. These year intervals allow us to examine how the impact of intercropping and phosphorus use efficiency changes over time, providing insight into the long-term effects of these practices. We also unified the term maize/soybean intercropping to “maize//soybean” Line 16,19,28,73,78, 80…so on.
To enhance clarity, we have checked the data calculation and experimental design in the manuscript and modified some indices calculations for example (can be found in Yield indices calculation).
Comment: Experimental Design Concerns
Response:
Regarding the reviewer’s concern about experimental design, we believe that the seven-year duration of the study provides significant value to understanding the long-term effects of continuous intercropping on maize yield and phosphorus use efficiency (PUE). We aimed to present the results in odd-year intervals (1st, 3rd, and 7th years) to provide a snapshot of the impact over time, as these represent key points in the progression of the experiment. This approach allows us to track short-term, mid-term, and long-term effects. However, we understand the concern and can include results from even-year intervals (2nd, 4th, 6th) in supplementary data (fig. B) for a more complete view.
|
Fig. B. Maize grain yield in fertilized and unfertilized plots under maize//soybean continuous intercropping in 2018, 2020, 2021, 2022. |
Comment: Data Calculation Issues
Response:
In the revised manuscript, we have clarified the methodology by adding related references for calculating indices such as Yield, Agronomic Efficiency (AE), Recovery Efficiency (RE), Partial Factor Productivity (PFP), and Contribution of Phosphate Fertilizer (CPF) and others in lines 390-397,399, 405-406, 410-414, 425-429, 440, 455.
Comment 1: Introduction: It is a feature of this study to discuss the effect of P fertilizer, except in the subtropical area, but the introduction part rarely introduces the research progress under P fertilizer intercropping, some more information needed. Line 73-74: not clear.
Response:
Thank you for your insightful comment. The introduction could benefit from a more detailed discussion of the research progress regarding phosphorus (P) fertilizer in intercropping systems, particularly in subtropical regions. We have expanded this section to provide more context on previous studies on P fertilizer use in intercropping and its implications for crop yield and sustainability. See lines (55 to 68)
Regarding lines 73-74, we revised in lines (88,89, and 92) the wording to ensure it is more comprehensible and provides a clearer explanation of the study’s focus.
Comment 2: Materials and Methods: There were too many complaints here. why no soybean monocropping design? why only analyze the 1, 3 and 7 years, but no 2, 4 and 6? odd year, not even year? the plot size was 4*6, and how the maize and soybean was arranged under such row spacing and plant spacing? Line 360: not clear, same amount N for monocropping and intercropping? the yield was air-dried for maize and soybean? Line 368: the illustration here was of some redundant, also the equation (1). Line 384: all the calculated indices need further checking, for the explanations and significance, and also calculation method. Line 400: the explanations was not clear and correct, here no repeat for monoculture or mixture, only year difference. Line 407: it seemed no unfertilized treatment here, and how the data came from? Line 430: the statistical analysis illustration was not clear, and SEM method not correct as shown in figure 4 because the index calculation was based on yield.
Comment: why no soybean monocropping design?
Response:
Thank you for your valuable comment. The omission of a soybean monocropping design was intentional and based on the primary objective of our study, which focused on evaluating the impacts of continuous maize/soybean intercropping on maize yield, phosphorus use efficiency, and sustainability. The aim was to assess how the combination of maize and soybean in an intercropping system would influence key variables, such as crop productivity and soil nutrient management when compared to a monocropped maize system. We specifically chose to compare maize monocropping with maize//soybean intercropping to highlight the distinct advantages that intercropping could offer over a mono-crop system, particularly in terms of resource use efficiency and crop yield stability. Including a soybean monocropping treatment was not deemed necessary for the focus of our analysis, as the intercropping system itself was the central variable of interest.
However, we acknowledge that soybean monocropping could provide additional insights into the dynamics of soybean growth and phosphorus use efficiency when grown alone. This could be explored in future studies to enhance the understanding of soybean's individual performance in relation to its intercropping benefits.
Comment: Why only analyze the 1, 3 and 7 years, but no 2, 4 and 6? odd year, not even year?
Response: analysis of 1, 3, and 7 years, but not 2, 4, and 6 years:
Thank you for your insightful question. The selection of the 1st, 3rd, and 7th years for analysis was based on our research objectives and the experimental design, to capture key intervals that reflect the short-term and long-term effects of intercropping. These specific years were selected to emphasize both the initial and cumulative impacts of intercropping.
Our primary objective was to evaluate how year-intervals of intercropping influence sustainability and phosphorus use efficiency, and we believed that the odd years (1st, 3rd, and 7th) would offer a comprehensive perspective on the changes in system performance over time, especially given the potential shifts in soil fertility and crop interactions as the system matured. Additionally, the choice of focusing on these odd years was intended to provide a strategic balance between early, intermediate, and later stages of the cropping system, while still ensuring statistical reliability and meaningful data capture across the timeline of the experiment. This allowed us to evaluate how the effects developed during the early years (1st year), as well as assess how the system responded over the intermediate years (3rd year), and its longer-term effects in the 7th year. However, we do have the data for the even years (2nd, 4th, and 6th), and we will provide this in the for clarification as illustrated in Fig B.
|
Fig. B. Maize grain yield in fertilized and unfertilized plots under maize//soybean continuous intercropping in 2018, 2020, 2021, 2022. |
Plot size and plant arrangement:
Response: The plot size 4*6= 24m2, which is quite enough to adjust maize and soybean rows according to the article (Zhou et al. (2024) as shown in line 365. Maize//Soybean Intercropping Improves Yield Stability and Sustainability in Red Soil under Different Phosphate Application Rates in Southwest China).
Comment: The same amount of nitrogen for monocropping and intercropping (Line 360) Response: Yes, the same amount of nitrogen was applied to both monocropping and intercropping treatments. Because our goal was to assess the effect of intercropping on nutrient use efficiency rather than different fertilization levels in line 380.
Comment: the yield was air-dried for maize and soybean?
Response: Thank you for your question. Yes, the yield was air-dried for both maize and soybean before further analysis. The plants were harvested and then left to dry at ambient conditions until they reached a consistent weight. This method was chosen to ensure that all moisture was removed, allowing us to accurately measure the dry biomass of the crops. Air-drying is a standard procedure in many agricultural studies, as it helps eliminate water content and provides a more consistent measure of the crop’s true biomass, free from moisture variability in lines 386-388.
Line 368: The illustration here was of some redundant, also the equation (1)
Response: Thank you for your comment regarding redundancy in the manuscript. We acknowledge that the explanation in the section may have been repetitive in lines 390-399. To improve clarity and readability, we have revised the content to streamline the explanation and removed redundant information. Specifically, we will focus on presenting the concept of the delta (Δ) symbol and its applications in the equations more concisely, ensuring that each term (such as Δ Yield, Δ SYI, Δ SI, etc.) is cleared without unnecessary repetition. We have also rechecked the relevance of Equation 1 which is also reported by Liu et al. (2023) in such a way.
Liu et al. (2023). Maize/soybean intercrop over time has higher yield stability relative to matched monoculture under different nitrogen-application rates. Field Crops Research, 301, 109015.
Zhou et al. (2024). Maize//Soybean Intercropping Improves Yield Stability and Sustainability in Red Soil under Different Phosphate Application Rates in Southwest China
Comment: Calculation of indexes and significance (Line 384)
Response:
Thank you for your comment. We have provided the actual standard formulas for each of the calculated indices, including the delta indices, in the "4.3. Yield indices calculation" section of the paper. This was done intentionally to ensure that readers have full transparency and understanding of how these indices were calculated.
The delta indices, such as delta yield, were calculated by subtracting the monocropping values from the intercropping values reported by Zhou et al. (2024). This method allows us to highlight the changes and differences between the two treatments, which helps in understanding the impact of intercropping on the various indices in lines 390-399.
For example, the formula for calculating the delta yield (Δ Yield) is as follows:
Δ Yield = Yield of Intercropping − Yield of Monocropping
Similarly, for other indices like the delta Sustainable Yield Index (Δ SYI), the delta Stability Index (Δ SI), the delta contribution rate of Phosphorus Fertilizer (Δ CRPF), and others, the same approach of subtraction was applied to capture the change between the intercropping and monocropping systems. We intentionally provided these formulas to give readers a standardized method to calculate and understand how the delta indices are derived and interpreted. We hope this clarifies the calculation method and the rationale behind providing the formulas for each index. ref. (Zhou et al. (2024). Maize//Soybean Intercropping Improves Yield Stability and Sustainability in Red Soil under Different Phosphate Application Rates in Southwest China)
Comment: The explanations was not clear and correct, here no repeat for monoculture or mixture, only year difference (Line 400):
Response: Line 400 was revised and clarified in Line 423-429, for easy understanding to avoid any misunderstanding.
Comment: Unfertilized treatment data (Line 407)
Response:
Thank you for your thoughtful comment. We apologize for any confusion regarding the unfertilized treatment. Let me clear that we did include an unfertilized treatment in our experiment, our main focus in the manuscript was on the changes in maize yield associated with intercropping compared to monocropping over year intervals. Therefore, we did not explicitly discuss the unfertilized treatment in the main text, as our primary objective was to highlight the yield differences between intercropping and monocropping. However, the unfertilized treatment data was used for calculating the Contribution of Phosphorus Fertilizer (CPF) in our study. To ensure transparency, we provided the full unfertilized treatment data in the Fig A as below. We hope this addresses your concern and clarifies the methodology used in the study.
|
|
|
Fig. A. Maize grain yield in fertilized and unfertilized plots under maize//soybean continuous intercropping |
Comment: Statistical analysis and SEM method (Line 430)
Response:
Thank you for your valuable comment. We applied Structural Equation Modeling (SEM) in our study to examine both the direct and indirect effects of various indices (such as AE, RE, PFP, SYI, SI, cropping pattern, and phosphorus fertilizer) on maize yield as reported by Liu et al. (2023). The use of SEM was crucial for understanding the complex relationships among these variables. Our SEM analysis revealed the direct and indirect impacts of these indices on maize yield. Additionally, the goodness-of-fit statistics, along with other model evaluation terms, confirmed that the SEM model was significant and robust, supporting the objectives of our research. We are confident that SEM provided a sound statistical framework for understanding the relationships in our data was revised in line 460-468.
(Liu et al. (2023). Maize/soybean intercrop over time has higher yield stability relative to matched monoculture under different nitrogen-application rates. Field Crops Research, 301, 109015).
Comment: Conclusion: not fully and comprehensively summarized, which should be mixture was appropriate for maize and soybean in stabilizing yield and nutrient use efficiency.
Response: Conclusion has been modified according to the reviewer’s suggestion in lines 470-483.
Other comments: The yield of the two crops under monoculture and mixture for each experimental year and the rainfall during the growth seasons for the seven years should be the results for the manuscript. Some were marked and annotated in the manuscript.
Response:
Thank you for your valuable comment. Our study aims to investigate the effect of continuous intercropping on the change in maize yield over year intervals. To achieve this, we calculated the delta yield (delat yield is defined as a symbol of relative quantity, which means the difference between the intercropping and monoculture involved yield Zhou et al., 2024) which represents the mean change in yield across the experimental years. This approach was central to our analysis, as it allows us to capture the variation in maize yield resulting from continuous intercropping over time. As such, we focused on the yield differences between intercropping and monocropping treatments over multiple years rather than presenting absolute yield values for each year. The delta yield provides a clearer picture of how the intercropping system impacts maize yield over time.
We understand the importance of including additional context like the rainfall and temperature data during the growth seasons, which may influence yield. Therefore, we have added the relevant rainfall and temperature data in the revised manuscript to provide a more complete picture of the factors affecting maize yield (in Figure 5 in line 383 to 384). In a similar study, Zhou et al. (2024) examined maize yield change under different fertilizer application rates over two years and used a delta yield approach to assess the effects. This reference might provide further clarity on how the delta yield approach has been successfully applied in yield stability and sustainability studies.
(Zhou et al. (2024). Maize//Soybean Intercropping Improves Yield Stability and Sustainability in Red Soil under Different Phosphate Application Rates in Southwest China).
We appreciate your suggestion and will make these revisions to enhance the clarity of our manuscript.
Annotated and marked comments in pdf file (Third reviewer):
Comment: (title) Because the maize and soybean were annual crops, thus planting years was not the key factor affecting its yield performance…, and the term year-interval is not accurate either.
Response: The title has been revised in lines 2-3.
Comment: the term “year-interval” was not accurate; in title
Response: In the title the term “year-term” has been changed. In terms of manuscript, so after carefully reviewing the use of this term in our manuscript, we believe that "year-interval" is an accurate description in the context of our study. The term was intended to refer to the specific intervals (or gaps) between the years in which data were collected, which, in our case, were the first year, third year, and seventh year of the experiment. These year-intervals were chosen to capture the changes in maize yield and phosphorus use efficiency over time, reflecting the effects of continuous intercropping and fertilization practices across different periods. The use of the term "year-interval" highlights the gaps between the data collection points (i.e., the first, third, and seventh years) and how these intervals represent the time evolution of the variables we were studying.
However, as we collected data over three distinct intervals, in some places in the manuscript, we use the plural form "year-intervals" when referring to all the intervals together. Both terms, "year-interval" and "year-intervals," are used consistently to describe these gaps, depending on the context (singular or plural).
We understand that the term might be interpreted differently by some readers, and we are open to further clarification if needed. We will ensure that the manuscript clearly explains the rationale behind the use of "year-intervals" or "year-interval" and revise accordingly for greater clarity.
Comment: Why no soybean intercropping treatment?
Response: Thank you for your comment. In our study, we set maize monocropping as the control because the primary focus was to evaluate the effects of intercropping on maize yield and phosphorus use efficiency (PUE). Since maize is the main crop of interest, using maize monocropping as the control allows us to assess the benefits of intercropping maize with soybean. While soybean monocropping could be helpful for a more comprehensive analysis, our research focused on maize. Thus, maize monocropping was selected as the control for comparison. We appreciate your suggestion and will consider it in future studies that may also focus on soybeans.
Comment: which should be mixture was appropriate for maize and soybean in stabilizing yield and nutrient use efficiency…
Response: Thank you for your comment regarding the appropriate mixture for maize and soybean in stabilizing yield and nutrient use efficiency. Based on the results of our experiment, we found that the maize-soybean intercropping system was effective in stabilizing yield and improving phosphorus use efficiency compared to monocropping systems.
Our study demonstrated that continuous intercropping with soybeans improved maize yield stability over the experimental years and increased phosphorus use efficiency. The intercropping system facilitated better nutrient cycling, reduced nutrient competition, and enhanced system sustainability, leading to higher phosphorus use efficiency across the different year intervals.
We also note that the specific planting configuration, including row spacing and plant density, which were detailed in the experimental design section of our manuscript, played a crucial role in optimizing the performance of the intercropping system. These factors should be considered when applying our findings to other contexts.
In conclusion, the maize-soybean intercropping system, with the row spacing and plant density used in our study, proved effective in stabilizing yields and improving nutrient use efficiency. The exact configuration may vary depending on local conditions, but our results demonstrate the benefits of this intercropping practice under the conditions tested.
We have revisited the conclusion section in the revised manuscript and improved the clarity and depth of our statements to better address the key findings of our study. (please check the conclusion)

Round 2
Reviewer 3 Report
Comments and Suggestions for Authors
The manuscript has been well revised and modified accordingly, and I think it can be accepted for publication. I SUGGEST the author to mention whether there is continuous cropping barrier for such system, and how to avoid or reduce the effect somewhere? The reference Zhou et al (2024) did not show the journal information.
Author Response
ponse to Reviewer’s Comments Comment : The results can be improved. The manuscript has been well revised and modified accordingly, and I think it can be accepted for publication. I SUGGEST the author mention whether there is continuous cropping barrier for such system, and how to avoid or reduce the effect somewhere? The reference Zhou et al (2024) did not show the journal information. Response: Thank you for your valuable suggestion. While our study focused on the effects of maize/soybean intercropping on maize yield, we acknowledge that continuous cropping can pose certain risks, such as nutrient depletion, pest buildup, and soil degradation. We have made some further improvements on the manuscripts in order to improve the same. The cropping boundaries have been captured on line 384-384. However, we did not observe significant barriers related to continuous cropping during the seven years of the experiment. The complementary nature of the maize/soybean intercropping system likely helped to mitigate some of the common challenges faced in monocropping systems, such as pest buildup and soil degradation. Moreover, we actively monitored soil health and applied appropriate fertilization practices to maintain soil fertility throughout the study. There are also potential barriers for continuous cropping in such systems. One such barrier is the unavailability of specific farm machinery, which could hinder the large-scale adoption of intercropping systems and affect productivity. Additionally, selecting specific maize and soybean varieties is essential to reduce competition between the crops, promote complementary interactions, and enhance the crops' resilience to unfavorable environmental conditions. For the references, harmonization has been done across with redundant ones being ironed out. In future work, we plan to explore additional strategies, such as incorporating organic amendments or enhancing soil health management practices, to further reduce the risks typically associated with continuous cropping.